# An interstellar synthesis of phosphorus oxoacids

Andrew M. Turner[1,2], Alexandre Bergantini [1,2], Matthew J. Abplanalp[1,2], Cheng Zhu [1,2], Sándor Góbi [1,2], Bing-Jian Sun[3], Kang-Heng Chao[3], Agnes H.H. Chang[3], Cornelia Meinert [4] & Ralf I. Kaiser [1,2]

Phosphorus signifies an essential element in molecular biology, yet given the limited solubility of phosphates on early Earth, alternative sources like meteoritic phosphides have been proposed to incorporate phosphorus into biomolecules under prebiotic terrestrial conditions. Here, we report on a previously overlooked source of prebiotic phosphorus from interstellar phosphine ($PH_3$) that produces key phosphorus oxoacids—phosphoric acid ($H_3PO_4$), phosphonic acid ($H_3PO_3$), and pyrophosphoric acid ($H_4P_2O_7$)—in interstellar analog ices exposed to ionizing radiation at temperatures as low as 5 K. Since the processed material of molecular clouds eventually enters circumstellar disks and is partially incorporated into planetesimals like proto Earth, an understanding of the facile synthesis of oxoacids is essential to untangle the origin of water-soluble prebiotic phosphorus compounds and how they might have been incorporated into organisms not only on Earth, but potentially in our universe as well.

[1] Department of Chemistry, University of Hawaii at Manoa, Honolulu, HI 96822, USA. [2] W.M. Keck Laboratory in Astrochemistry, University of Hawaii at Manoa, Honolulu, HI 96822, USA. [3] Department of Chemistry, National Dong Hwa University, Shoufeng, 974 Hualien, Taiwan. [4] Université Côte d'Azur, CNRS, Institut de Chimie de Nice, Nice, France. Correspondence and requests for materials should be addressed to R.I.K. (email: ralfk@hawaii.edu)

The ubiquitous presence of the phosphorus (V) oxidation state in contemporary biomolecules found in phospholipids, ADP/ATP, and RNA/DNA represents a hitherto unresolved "phosphorus problem" as the deprived solubility of phosphate minerals such as apatite ($Ca_5(PO_4)_3(F,Cl,OH)$) limits bioavailable phosphorus for the first organisms on early Earth[1,2]. Efficient prebiotic mechanisms toward the formation of high-energy phosphates such as polyphosphates containing the repeating $(HPO_3)_n$ moiety and phosphate esters have remained elusive as well[3]. Phosphate diesters constitute the backbone of RNA and DNA—key molecules carrying the genetic information for the reproduction of all known living organisms—as well as the hydrophilic head of phospholipids, while monoesters of pyrophosphate ($P_2O_7^{4-}$) and triphosphate ($P_3O_{10}^{5-}$) play a critical role in cellular energy transfer as adenosine diphosphate (ADP) and triphosphate (ATP), respectively. To overcome this "phosphorus problem", soluble molecules carrying the phosphorus (III) oxidation state have been proposed with the alkylphosphonic acids ($RP(O)(OH)_2$; R = methyl, ethyl, propyl, butyl) detected in the Murchison meteorite serving as soluble compounds of extraterrestrial origin[4]. Prebiotic phosphorus chemistry could have been initiated by phosphorus (III) or the first organisms might have oxidized bioavailable phosphorus (III) to (V). Yet, the underlying synthetic routes to these phosphorus-bearing biomolecules along with their precursors are still in their infancy. This fundamental knowledge on the origins of the phosphorus chemistry is critical to unravel how the phosphorus biochemistry and life itself might have emerged on early Earth.

In this communication, we reveal a facile and versatile pathway toward the abiotic synthesis of phosphorus (III) and (V) oxoacids in phosphine ($PH_3$)-doped interstellar analog ices of water ($H_2O$) and carbon dioxide ($CO_2$) upon interaction with ionizing radiation in the form of energetic electrons, which simulate secondary electrons generated in the track of galactic cosmic rays penetrating interstellar ices at temperatures as low as 5 K[5]. Phosphine has been observed in extraterrestrial environments such as in the circumstellar envelope of the carbon star IRC + 10216[6] along with other phosphorus-containing compounds such as CP[7], CCP[8], HCP[9], PN[10–12], and NCCP[13], and also within our Solar System in the atmospheres of Jupiter and Saturn[14,15]. More recently, phosphine was attributed as the source of the phosphorus signal in comet 67P/Churyumov-Gerasimenko[16]. Surface reactions involving schreibersite, an iron–nickel phosphide that along with phosphates compose the bulk of phosphorus in meteorites[17], in aqueous solutions resembling conditions of early Earth have been suggested, but do not fully reproduce the synthesis of alkylphosphonic acids[18–22], although the phosphorylation of adenosine has been demonstrated[23]. Thus, the formation of biorelevant, phosphorus-bearing molecules on interstellar grains—silicate-based nanoparticles coated with a few hundred nanometers of water ($H_2O$), methanol ($CH_3OH$), carbon monoxide (CO), carbon dioxide ($CO_2$), methane ($CH_4$), formaldehyde ($H_2CO$), and ammonia ($NH_3$) ices[24] from which interstellar phosphine would be expected to be depleted onto from the gas phase—in molecular clouds may represent a key process leading to a facile synthesis of phosphorus compounds of potential biological significance. Considering that molecular clouds constitute the nurseries of stars and planetary systems—including our own—the identification of phosphonic acid ($H_3PO_3$), phosphoric acid ($H_3PO_4$), and pyrophosphoric acid ($H_4P_2O_7$) in our ices suggests that these phosphorus oxoacids might have been at least partially incorporated into our Solar System from interstellar matter via circumstellar disks. Further, these compounds could have been delivered to early Earth by comets or meteorites such as Murchison as sources of phosphorus for biomolecules with both single phosphate groups (RNA/DNA and phospholipids) and phosphate chains (ADP/ATP). Our study defines the molecular complexity of phosphorus-bearing molecules synthesized in interstellar ices and unveils where in the universe discrete, phosphorus-containing molecular precursors relevant to the origins of life might have been synthesized and even incorporated into contemporary biomolecules.

## Results

**Infrared spectroscopy**. In the infrared spectra of irradiated, phosphine-bearing ices, critical functional groups of phosphorus oxoacids were identified at 5 K and also in the residues that remained after annealing the exposed ices to 300 K (Supplementary Tables 1 and 2)[25]. These bands include stretching modes of phosphorus–oxygen single and double bonds [$\nu(P-O)$ (800–950 cm$^{-1}$), $\nu(P=O)$ (1140–1300 cm$^{-1}$)], the deformation mode of the OPOH functional group [$\delta(O=P-OH)$ (1550–1710 cm$^{-1}$)], and the P–OH moiety [$\nu$(O-H) (2170 and 2700 cm$^{-1}$)]. These findings suggest that functional groups linked to oxoacids of phosphorus are the results of an exposure of the ices at temperatures as low as 5 K. Experiments with $^{18}O$-labeled ice constituents ($H_2^{18}O$ and $C^{18}O_2$) match the isotopic shifts determined for redshifted $\nu(P-O)$ and $\nu(P=O)$ modes by ~30 cm$^{-1}$[26,27]. Therefore, the infrared analysis reveals the existence of functional groups [P-O, P=O, and O=P-OH] present in phosphorus oxoacids; however, considering that fundamentals and hence the absorptions of functional of oxoacids, such as phosphonic and phosphoric acid, fall in the same range, infrared spectroscopy does not allow an identification of individual oxoacids nor their isomers. In other words, functional groups are not unique to specific phosphorus oxoacids and hence do not identify any particular molecule. Therefore, alternative analytical techniques are critical.

**PI-ReTOF-MS**. To probe discrete molecular species, we exploited photoionization reflectron time-of-flight mass spectrometry (PI-ReTOF-MS) during the annealing of the irradiated ices. This method represents a unique approach to identify gas phase molecules isomer-selectively after photoionization based on their distinct ionization energies[28]. Here, the subliming products were photoionized in separate experiments with 10.86 and 9.93 eV photons to elucidate the nature of the oxoacid isomer(s) synthesized. Accounting for the adiabatic ionization energies (IE) of the oxoacids ($H_3PO_x$; $x$ = 1–4) (Table 1), the energy of the 10.86 eV photon is above the IE of each oxoacid; therefore, if formed and if subliming, all oxoacids can be ionized at this photoionization energy. The use of PI-ReTOF-MS necessitated isotopically labeled $H_2^{18}O$ and $C^{18}O_2$ reagents to distinguish between products formed at identical masses. Phosphinic acid and hypophosphorous acid ($H_3PO_2$) along with diphosphine ($P_2H_4$) have a molecular weight of 66 amu; on the other hand, $^{18}O$-substituted phosphinic acid and hypophosphorous acid ($H_3P^{18}O_2$) shifts the mass by 4 to 70 amu, thus discriminating these acids from diphosphine. At the photoionization energy of 10.86 eV, the temperature-programmed desorption (TPD) profiles indicate that $H_3P^{18}O$ and $H_3P^{18}O_2$ oxoacids, which are associated with signal at $m/z = 52$ and 70, respectively, are formed in both $PH_3$-$H_2^{18}O$ and $PH_3$-$C^{18}O_2$ systems (Fig. 1). As the irradiation current increases from 100 nA via 1000–5000 nA, the ion counts at $m/z = 52$ ($H_3P^{18}O^+$) decrease by a factor of about eight in the $PH_3$-$H_2^{18}O$ system; simultaneously, the signal at $m/z = 70$ ($H_3P^{18}O_2^+$), which is absent at 100 nA, arises. This finding may suggest that phosphinic/hypophosphorous acid ($H_3P^{18}O_2$) is formed from phosphine oxide ($H_3P^{18}O$) likely via reaction with atomic oxygen. The analysis of the $PH_3$-$C^{18}O_2$ systems supports this conclusion: phosphinic/hypophosphorous acid ($H_3P^{18}O_2$; 70 amu) is only detected at higher irradiation currents of 1000 and 5000 nA. On the other hand, PI-ReTOF-MS did not succeed in the detection of phosphorous/phosphonic acid ($H_3P^{18}O_3$; $P(^{18}OH)_3$/$HP^{18}O(^{18}OH)_2$) or phosphoric acid ($H_3P^{18}O_4$) due to their low volatility that limits their sublimation from the substrate.

**Table 1 Calculated adiabatic ionization energies and relative energies of various phosphorus oxoacids**

| Structure | Formula | Name | Ionization Energy (eV)[a] | Relative Energy (eV) |
|---|---|---|---|---|
| | $H_3PO$ | Phosphine oxide | 10.71 | 0.00 |
| | cis-$H_2POH$ | cis-Hydroxyphosphine | 9.22 | 0.03 |
| | trans-$H_2POH$ | trans-Hydroxyphosphine | 9.23 | 0.03 |
| | $H_2P(O)OH$ | Phosphinic acid | 10.71 | 0.00 |
| | $HP(OH)_2$ | Hypophosphorous acid | 8.94 | 0.37 |
| | $HPO(OH)_2$ | Phosphonic acid | 10.83 | 0.00 |
| | $P(OH)_3$ | Phosphorous acid | 8.68 | 0.52 |
| | $PO(OH)_3$ | Phosphoric acid[b] | 10.69 | 0.00 |

[a]Ionization potential by CCSD(T)/CBS with B3LYP/cc-pVTZ zero-point energy correction
[b]In the ice, the $C_3$ symmetric phosphoric acid exists as two enantiomers:

Considering the signals detected at $m/z = 52$ ($H_3P^{18}O^+$) and 70 ($H_3P^{18}O_2^+$) in the 10.86 eV experiment, we were interested in untangling the nature of the structural isomer(s) formed. Since the ionization energies of the $H_3P^{18}O$-$H_2P^{18}OH$ and $H_2P(^{18}O)$($^{18}OH$)-$HP(^{18}OH)_2$ isomer pairs are separated by more than 1.5 eV (Table 1), a second set of experiments was carried out at a photoionization energy of 9.93 eV. This energy is below the ionization energies of the phosphine oxide ($H_3P^{18}O$) and phosphinic acid ($H_2P(^{18}O)^{18}OH$) isomers, but above the ionization energies of the hydroxyphosphine ($H_2P^{18}OH$) and hypophosphorous acid ($HP(^{18}OH)_2$) isomers. A close inspection of the TPD profiles of $m/z = 52$ (Fig. 1) reveals that the TPD profiles are nearly identical at 10.86 and 9.93 eV, suggesting that

at least the thermodynamically less stable hydroxyphosphine isomer ($H_2P^{18}OH$) is formed; since the absolute photoionization cross-sections of both isomers are unknown, the presence of phosphine oxide cannot be proven. However, Withnall and Andrews explored in previous matrix isolation experiments the chemistry of phosphine−molecular oxygen samples[26,27] and revealed the formation of hydroxyphosphine ($H_2POH$) with smaller contributions of the phosphine oxide isomer ($H_3PO$). Finally, we compare the TPD profiles of $m/z = 70$ recorded at 10.86 and 9.93 eV. Upon lowering the photon energy to 9.93 eV, the signal at $m/z = 70$ vanishes; therefore, we can conclude that only the thermodynamically preferred phosphinic acid isomer ($H_2P(^{18}O)^{18}OH$) is formed, but no hypophosphorous acid ($HP(^{18}OH)_2$). The higher molecular weight of phosphinic acid (70 amu) compared to hydroxyphosphine (52 amu) is also associated with an increase of the sublimation temperature in the range of 260–300 K in contrast to 160–240 K.

**TOF-SIMS.** Having established the synthesis of at least two of the simplest phosphorus oxoacids (hydroxyphosphine and phosphinic acid) and possibly phosphine oxide by exploiting PI-ReTOF-MS, we searched for higher molecular-weight oxoacids in the residues of the annealed samples utilizing time-of-flight secondary ion mass spectrometry (TOF-SIMS). TOF-SIMS facilitates the sputtering of the solid residues and detects the ions in both positive and negative ion detection modes (Fig. 2). Since the sputtering might also fragment the phosphorus oxoacids, these fragmentation patterns have to be determined. This assists in an identification of well-defined mass-to-charge ratios unique to each of the oxoacids and also allows a quantification of the oxoacids formed. The results of the calibration of phosphonic acid ($H_3PO_3$), phosphoric acid ($H_3PO_4$), and pyrophosphoric acid ($H_4P_2O_7$) are compiled in Supplementary Tables 3 and 4. The negative ion spectra are very sensitive to probe the oxoacids via their deprotonated parent molecules. Here, pyrophosphoric ($H_4P_2^{18}O_7$) and phosphonic acid ($H_3P^{18}O_3$) can be identified in all residues via their unique signals of $HP_2^{18}O_6^-$ / $H_3P_2^{18}O_7^-$ and $H_2P^{18}O_3^-$, respectively. While pyrophosphoric acid has a fragment of small intensity for $H_2P^{18}O_4^-$, the low quantity of $H_4P_2^{18}O_7$ in our residues contributes a minor amount to the moderately intense $H_2P^{18}O_4^-$ signal, which can be attributed to phosphoric acid ($H_3P^{18}O_4$). As a general trend, the yield of each of these oxoacids increases with the irradiation dose; significantly enhanced yields are seen in carbon dioxide bearing ices compared to water-rich ices especially at higher doses. Although phosphinic acid is subliming at 260–300 K as verified in the PI-ReTOF-MS analysis, the SIMS analysis revealed a strong peak for $H_2P^{18}O_2^-$; this ion is not observed as a fragment from any calibration compound, but can be formally linked to phosphinic acid ($H_3P^{18}O_2$). A close look at the PI-ReTOF-MS data (Fig. 1) indicates that the intensity of $m/z = 70$ ($H_3P^{18}O_2^+$) does not completely lead to zero at 300 K; therefore, a fraction of phosphinic acid ($H_3P^{18}O_2$) is likely to reside in the solid residue. Although holding a lower sensitivity, the positive ion spectra confirm the assignments derived from the negative ion mode. Pyrophosphoric acid ($H_4P_2^{18}O_7$), phosphonic acid ($H_3P^{18}O_3$), and phosphoric acid ($H_3P^{18}O_4$) could be detected via their protonated counterparts, i.e., $H_5P_2^{18}O_7^+$, $H_4P^{18}O_3^+$, and $H_4P^{18}O_4^+$.

**Gas chromatography.** Finally, the phosphorus oxoacids in the residues were also extracted, derivatized as trimethylsilyl (TMS) esters (–$OSi(CH_3)_3$), and analyzed via two-dimensional gas chromatography time-of-flight mass spectrometry. A TOF-MS was exploited to record the retention times along with the mass spectra (Supplementary Table 5). This protocol led to the detection of three

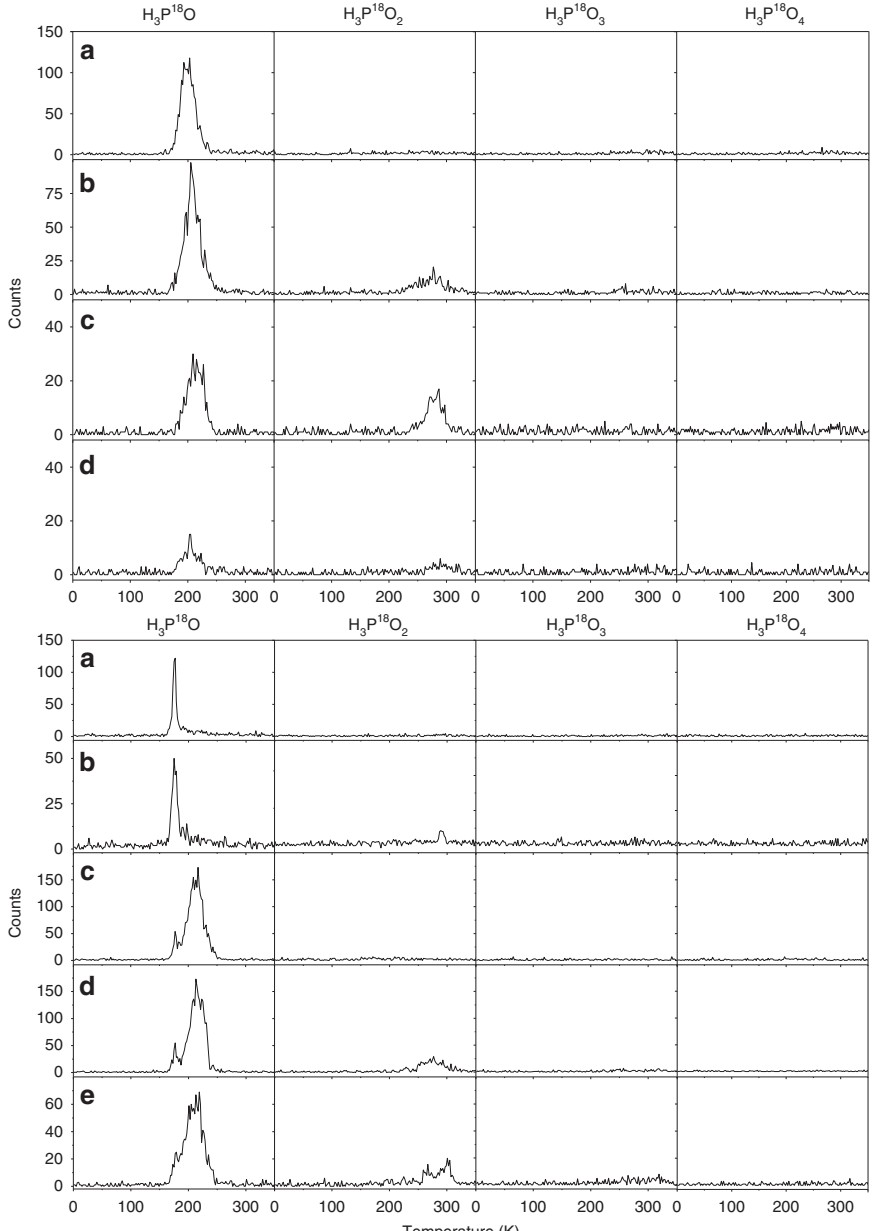

**Fig. 1** PI-ReTOF-MS data showing the temperature-programmed desorption profiles for phosphorus oxoacids. Each column displays the profiles for $m/z = 52$ ($H_3P^{18}O$), $m/z = 70$ ($H_3P^{18}O_2$), $m/z = 88$ ($H_3P^{18}O_3$), and $m/z = 106$ ($H_3P^{18}O_4$). Top: Ices of phosphine ($PH_3$)–carbon dioxide ($C^{18}O_2$) at irradiation currents of **a** 100 nA, 9.93 eV photoionization energy, **b** 100 nA, 10.86 eV photoionization energy, **c** 1000 nA, 10.86 eV photoionization energy, and **d** 5000 nA, 10.86 eV photoionization energy. Bottom: Ices of phosphine ($PH_3$)–water ($H_2^{18}O$) at irradiation currents of **a** 100 nA, 9.93 eV photoionization energy, **b** 100 nA, 10.86 eV photoionization energy, **c** 1000 nA, 9.93 eV photoionization energy, **d** 1000 nA, 10.86 eV photoionization energy, and **e** 5000 nA, 10.86 eV photoionization energy. The PI-ReTOF-MS data for the blank experiments are shown in Supplementary Fig. 2

phosphorus oxoacids (Fig. 3): phosphoric acid ($H_3P^{18}O_4$), phosphonic acid ($HP^{18}O(^{18}OH)_2$), and phosphinic acid ($H_2P^{18}O(^{18}OH)$). Phosphoric acid could be identified via the molecular ion of the tris (trimethylsilyl)ester ($^{18}OP(^{18}Osi(CH_3)_3)_3$) at $m/z$ ($M^+$) = 322 and its fragment originating from the loss of a methyl group at $m/z$ ($M^+ -15$) = 307[29]. Phosphonic acid was detected via the molecular ion of the derivatized phosphorous acid tautomer in the form of its tris (trimethylsilyl)ester ($P(^{18}OSi(CH_3)_3)_3$) at $m/z$ ($M^+$) = 304 and its fragment of the methyl group loss at $m/z$ ($M^+ -15$) = 304[30]. Finally, phosphinic acid could be sampled via its hypophosphorous acid tautomer as its bis(trimethylsilyl)ester ($HP(^{18}OSi(CH_3)_3)_2$) at $m/z$ ($M^+$) = 214 and also by its fragment of the methyl group elimination at $m/z$ ($M^+ -15$) = 199. Derivatization as trimethylsilyl

(TMS) esters shifts the tautomeric phosphonic–phosphorous and phosphinic–hypophosphorous acid equilibrium to the phosphorous and hypophosphorous acid esters (Supplementary Fig. 1)[31]. Calibration experiments suggest that the TMS derivative of pyrophosphoric acid as detected in small quantities via SIMS was found to be thermally unstable and decomposed on the GC columns[32]. Consequently, with the exception of pyrophosphoric acid, the GC×GC-TOF-MS analysis correlates exceptionally well with the SIMS data that detected key phosphorus oxoacids in the residues of the irradiated phosphine-doped interstellar analog ices.

**Discussion.** Having identified four monophosphorus oxoacids [hydroxyphosphine ($H_3PO$: $PH_2OH(-I)$), phosphinic acid

($H_3PO_2$: $H_2P(O)OH(+I)$), phosphonic acid ($H_3PO_3$: $HP(O)(OH)_2(+III)$), phosphoric acid (phosphoric acid ($H_3PO_4(+V)$)] along with pyrophosphoric acid ($H_4P_2O_7(+V)$) with phosphorus in four distinct oxidation states ranging from –I to +V, we are discussing now possible formation pathways. For simplicity in the following discussion, the $^{18}O$ label is dropped. It should be noted that although the FTIR analysis provided evidence on the emergence of functional groups associated with phosphorus oxoacids even at 5 K, the FTIR data were unable to identify individual oxoacids due to overlapping absorptions of the functional groups. Therefore, kinetic profiles linked to the formation of individual phosphorus oxoacids could not be provided. However, a few important conclusions can be drawn. First, based on the PI-ReTOF-MS data recorded at 100, 1000, and 5000 nA, the yields of $PH_2OH(-I)$ and $H_2P(O)OH(+I)$ depend on the irradiation current and hence dose. Recall that in the $PH_3$–$H_2O$ system, $H_2P(O)OH(+I)$ is absent in the 100 nA experiment, but emerges at 1000 nA. This observation suggests a sequential formation of phosphorus oxoacids via stepwise reaction of oxygen atoms starting with phosphine ($PH_3$). Considering the water- and carbon dioxide-rich ices, the radiolysis of water and carbon dioxide can generate electronically excited oxygen atoms in strongly endoergic reactions[5,33–35]; water can also decompose via the formation of atomic hydrogen and hydroxyl radicals (OH) (Eq. 1–3). The required energy for the bond dissociation can be supplied by the energetic electrons.

$$CO_2(X^1\Sigma_g^+) \rightarrow CO(X^1\Sigma^+) + O(^1D) \quad \Delta_R G = 7.59\,eV \quad (1)$$

$$H_2O(X^1A_1) \rightarrow H(^2S) + OH(X^2\Pi_\Omega) \quad \Delta_R G = 4.83\,eV \quad (2)$$

$$H_2O(X^1A_1) \rightarrow H_2(X^1\Sigma_g^+) + O(^1D) \quad \Delta_R G = 6.74\,eV \quad (3)$$

Simultaneously, the energetic electrons can also lead to a phosphorus–hydrogen bond rupture in phosphine leading to phosphino ($PH_2$) radicals (Eq. 4)[36].

$$PH_3(X^1A_1) \rightarrow PH_2(X^2B_1) + H(^2S) \quad \Delta_R G = 3.51\,eV \quad (4)$$

In the water–phosphine system, the formation of $PH_2OH$ may proceed via a barrier-less recombination of the phosphino ($PH_2$) radical with the hydroxyl (OH) radical (Eq. 5). Alternatively, electronically excited oxygen atoms can insert without barrier into the phosphorus–hydrogen bond leading to $PH_2OH$ as well (Eq. 6). It is important to note that electronically excited oxygen atoms can also add without barrier to the phosphorus atom of phosphine yielding phosphine oxide ($H_3PO$) (Eq. 7), which can then undergo hydrogen migration to form $PH_2OH$[26,27,37]. Schmidt et al. suggested that a barrier between 3.01 and 3.73 eV separated both isomers[38]; this energy can be supplied by the energetic electrons as well. Recall that the formation of phosphine oxide could not be proven or disproven in our study. We would like to stress that Stief et al. also probed the gas phase kinetics of ground-state atomic oxygen with phosphine over a temperature range of 208–423 K revealing that ground-state oxygen preferentially adds to the phosphorus atom[39,40]; unfortunately, neither products were identified, nor the role of intersystem crossing

from the triplet to the singlet surface were probed.

$$PH_2(X^2B_1) + OH(X^2\Pi_\Omega) \rightarrow PH_2OH(X^1A') \quad \Delta_R G = -5.27\,eV \quad (5)$$

$$PH_3(X^1A_1) + O(^1D) \rightarrow PH_2OH(X^1A') \quad \Delta_R G = -8.27\,eV \quad (6)$$

$$H_3PO(X^1A_1) \rightarrow PH_2OH(X^1A') \quad \Delta_R G = 0.03\,eV \quad (7)$$

Once $PH_2OH$ is formed, the addition of another electronically excited oxygen atom to the phosphorus atom produces $H_2P(O)OH$ (Eq. 8). Successive insertions of electronically excited oxygen atoms in phosphorus–hydrogen bonds may lead via $HP(O)(OH)_2$ (Eq. 9) to $H_3PO_4$ (Eq. 10). Essentially, in this reaction sequence, up to four oxygen atoms are required to oxidize phosphine to ultimately phosphoric acid via stepwise oxidation. The release of up to four oxygen atoms requires 27.0 eV and 30.4 eV to be generated from water and carbon dioxide, respectively. Therefore, thermal reactions cannot lead to the oxoacids at 5 K, but cosmic-ray-triggered non-equilibrium chemistry is required to supply the required oxygen atoms (and possibly the hydroxyl radicals) for the oxidation process.

$$PH_2OH(X^1A') + O(^1D) \rightarrow$$
$$H_2P(O)OH(X^1A') \quad \Delta_R G = -7.68\,eV \quad (8)$$

$$H_2P(O)OH(X^1A') + O(^1D) \rightarrow$$
$$H_2P(O)(OH)_2(X^1A') \quad \Delta_R G = -7.81\,eV \quad (9)$$

$$H_2P(O)(OH)_2(X^1A') + O(^1D) \rightarrow$$
$$H_3PO_4(X^1A) \quad \Delta_R G = -7.62\,eV \quad (10)$$

In regard to the conversion yields, the infrared analysis indicates that $76 \pm 8\%$ of the phosphine reacted in the 10:1 $CO_2$:$PH_3$ ice; this was determined by utilizing the comparative area of the $\nu_2$ and $\nu_4$ bands of phosphine. Similarly, the $\nu_2$, $2\nu_2 + \nu_3$, and $\nu_1 + \nu_3$ bands of carbon dioxide show that $28 \pm 1\%$ of $CO_2$ was destroyed after the irradiation. This equates to $8 \pm 2 \times 10^{16}$ molecules of phosphine and $3 \pm 1 \times 10^{17}$ molecules of carbon dioxide destroyed by irradiation. Since each irradiated carbon dioxide molecule liberates only one oxygen atom, the ratio of available reactive phosphorus-to-oxygen atoms is thus $1.0:3.4 \pm 0.8$. If all of the reacted phosphine became incorporated into one of the three simplest oxoacids, the $8 \pm 2 \times 10^{16}$ molecules produced suggest that $2.1 \pm 0.8 \times 10^{-4}$ molecules were formed per eV of irradiation. The formation of $H_3PO_4$ is limited by the number of oxygen atoms generated, and at most $7 \pm 2 \times 10^{16}$ molecules of $H_3PO_4$ could form, which corresponds to a generation of $1.7 \pm 0.6 \times 10^{-4}$ $H_3PO_4$ molecules $eV^{-1}$. The $\nu_2$ and $\nu_L$ bands of water were analyzed for the 10:1 $H_2O$:$PH_3$ ice, and $62 \pm 2\%$ of phosphine was found to have reacted, while only $37 \pm 5\%$ of the water was destroyed. This is equivalent to $6 \pm 2 \times 10^{16}$ phosphine molecules and $8 \pm 3 \times 10^{17}$ water molecules. In this case, much more oxygen was available compared to phosphorus, with a 1:12 phosphorus-to-oxygen atom ratio, and thus phosphorus will always be the limiting atom in the formation of the oxoacids. Here, the upper yields of $6 \pm 2 \times 10^{16}$ molecules of phosphorus oxoacids are formed at $1.6 \pm 0.6 \times 10^{-4}$ molecules $eV^{-1}$.

Using reagent standards, the quantity of phosphoric acid ($H_3PO_4$) in the residues can be determined via GC×GC-TOF-MS and compared to the infrared spectra to establish a reaction yield.

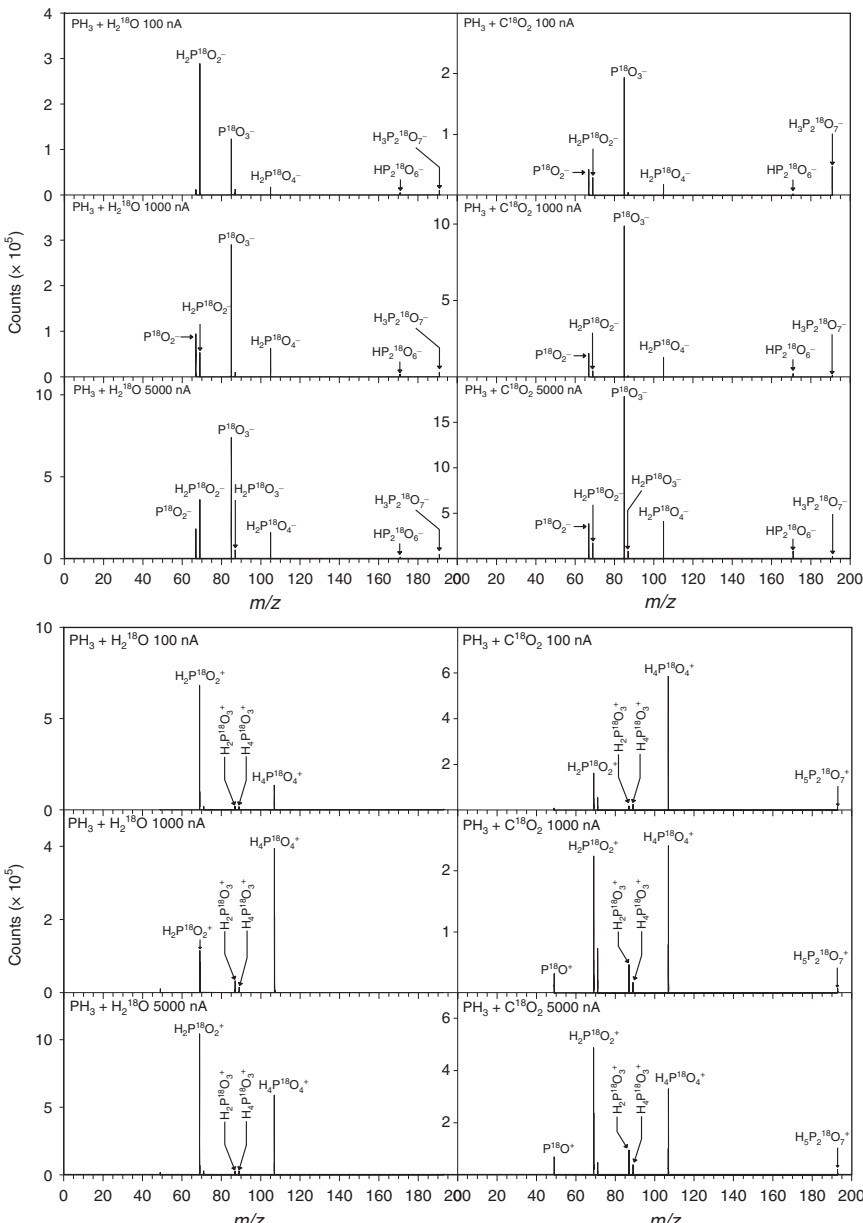

**Fig. 2** SIMS data from residues of irradiated phosphine-doped ices. The spectra were recorded in the negative (top) and positive ion mode (bottom) correlated with the formation of $^{18}O$-substituted oxoacids formed in irradiated phosphine ($PH_3$)–water ($H_2^{18}O$) (right) and phosphine ($PH_3$)–carbon dioxide ($C^{18}O_2$) (left) ices

The $CO_2$:$PH_3$ ices produced 5 nmol ($3 \times 10^{15}$ molecules) of phosphoric acid, while only 1 nmol ($6 \times 10^{14}$ molecules) was detected from the $H_2O$:$PH_3$ ice, which results in $7 \pm 3 \times 10^{-6}$ molecules of $H_3PO_4$ per eV in the $CO_2$:$PH_3$ ice and $2 \pm 1 \times 10^{-6}$ $H_3PO_4$ molecules $eV^{-1}$ in the $H_2O$:$PH_3$ ice. Thus, phosphoric acid represents 4% of the reacted phosphine in $CO_2$:$PH_3$, while a 1% yield was found in the $H_2O$:$PH_3$ system. As phosphoric acid was the most abundant compound detected in the residues, this indicates that most of the phosphorus sublimed during the TPD, for example, as diphosphane as detected experimentally.

In conclusion, by exposing phosphine ($PH_3$)-doped interstellar analog ices to ionization radiation and exploiting an array of complementary in situ and ex situ analytical tools, the present study offers compelling evidence on a facile formation of distinct oxoacids of phosphorus: phosphinic acid ($H_3PO_2$) P(I), phosphonic acid ($H_3PO_3$) P(III), phosphoric acid ($H_3PO_4$) P(V), and pyrophosphoric acid ($H_4P_2O_7$) P(V). The formation of those

oxoacids can be initiated via a radical–radical recombination between the hydroxy (OH) and phosphino ($PH_2$) radicals (Eq. 5) or through insertion of electronically excited atomic oxygen, released by unimolecular decomposition of water and carbon dioxide, with phosphine-forming phosphinic acid ($PH_2OH$) (Eq. 6). Once phosphinic acid ($PH_2OH$) forms, successive addition and insertion of electronically excited oxygen atom to the phosphorus atom produce phosphonic acid ($H_3PO_3$) and phosphoric acid ($H_3PO_4$) (reactions 8–10). Overall, up to four oxygen atoms are needed to oxidize phosphine to phosphoric acid; this requires up to about 30 eV. Hence, thermal reactions cannot form oxoacids at 5 K, but cosmic-ray-triggered non-equilibrium chemistry is required to supply the required oxygen atoms (and possibly the hydroxyl radicals) for the oxidation process. Please note that our studies were carried out in the condensed (ice) phase, but not under single collision conditions in the gas phase[41]. Therefore, it is not feasible neither in the

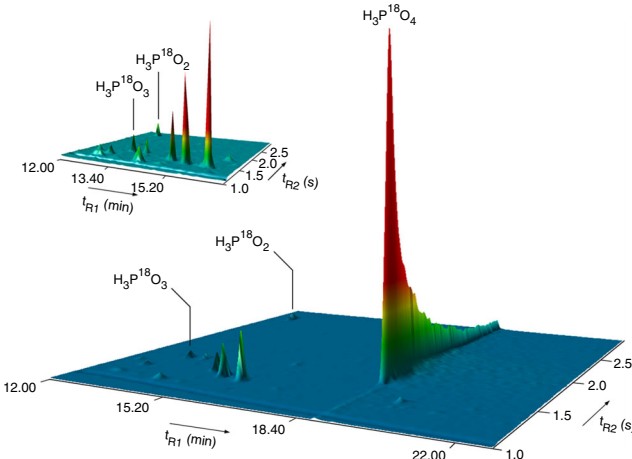

**Fig. 3** Multidimensional gas chromatogram showing $^{18}$O-labeled phosphorus oxoacids extracted from the residues. The atomic mass units 214 (×100) and 304 (×150) were selected for this multidimensional chromatographic representation. Partial GC×GC chromatogram of the separation of the two minor phosphorus oxoacids is shown top left. The unassigned peaks result from the silylation agent (BSTFA: TMCS 1%) that was injected in excess compared to the blank analysis (Supplementary Fig. 3) to avoid sample loss

present studies nor in any other laboratory to determine the efficiency of each elementary reaction (oxidation step) involved in the formation of the individual oxoacids. This would require pulse-probe experiments with femtosecond (few 10 fs pulses) electron pulses penetrating the ice sample. These experiments do not exist yet.

Whereas on Earth, phosphine is classified as highly toxic and only slightly soluble in water, the present work reveals interstellar phosphine as a critical precursor in the synthesis of highly water-soluble phosphorus oxoacids prevalent in contemporary biochemistry. Phosphoric acid ($H_3PO_4$) in particular presents a soluble source of phosphorus in the phosphorus (V) oxidation state as found in RNA/DNA and ADP/ATP. The identification of pyrophosphoric acid ($H_4P_2O_7$)—a formal condensation product of phosphoric acid—is significant since our investigations expose that polyphosphates such as diphosphates found in ADP can be formed in interstellar ices upon interaction with ionizing radiation mimicking typical life times of molecular clouds of a few $10^6$ years. The unsuccessful detection of triphosphoric acid ($H_5P_3O_{10}$)—a precursor to the phosphorus backbone of ATP—could indicate that the phosphorus chemistry in interstellar ices ceases with the synthesis of pyrophosphoric acid over the lifetime of a molecular cloud, thus defining the molecular complexity of phosphorus-bearing oxoacids formed in extraterrestrial environments.

Finally, the synthesis of phosphonic acid ($H_3PO_3$) could support an alternative route to phosphorus (III) as found in alkylphosphonic acids in the Murchison meteorite. As these alkylphosphonic acids are of continued interest to the "Origins of Life" community, the incorporation of methane ($CH_4$) or more complex hydrocarbons to interstellar ices has the potential to yield a complex mixture of alkylphosphonic acids as detected in Murchison and, with derivatives of phosphoric acid, biologically relevant phosphate esters. While other interstellar phosphorus-bearing molecules, such CP or HCP[42], have been considered as a source of phosphorus for alkylphosphonic acids via gas or aqueous phase reactions, phosphine ($PH_3$) represents the ideal compound for phosphorus-based interstellar ice experiments, which has been a missing piece in the astrochemical-phosphorus literature that is rich with gas and mineral phase studies. The

phosphorus oxoacids detected in our experiments might have also been formed within the ices of comets such as 67P/Churyumov-Gerasimenko, whose phosphorus source is believed to derive from phosphine[16]. Since comets contain at least partially the remnants of the material of the protoplanetary disk that formed our Solar System, these compounds might be traced back to the interstellar medium wherever sufficient phosphine in interstellar ices is available. Upon delivery to Earth by meteorites or comets, these phosphorus oxoacids would have been available for Earth's prebiotic phosphorus chemistry. For example, recent studies have shown phosphoric acid[43] and diamidophosphate—a possible derivative of trimetaphosphate[44]—can abiotically phosphorylate various prebiotic compounds such as sugars, amino acids, and nucleotides in aqueous solution to produce higher-order biologically relevant molecules. Although future studies have to be conducted on the underlying formation mechanisms of these oxoacids, the present study embodies a step toward elucidating possible abiotic pathways toward oxoacids of phosphorus resembling key molecular building blocks in contemporary biochemistry on Earth thus bringing us closer to eventually predicting where in the galaxy molecular precursors linked to the origins of life might have been synthesized.

## Methods

**Experimental**. Ices of phosphine (99.9995%), carbon dioxide (99.999%), and water (HPLC grade), along with oxygen-18-labeled carbon dioxide (95 atom % $^{18}$O) and water (99 atom % $^{18}$O), were deposited onto a polished silver wafer. This silver substrate was mounted onto an oxygen-free high-conductivity copper cold head capable of achieving temperatures as low as 5 K by a closed-cycle helium refrigerator (Sumitomo Heavy Industries, RDK-415E) inside a contamination-free stainless steel chamber capable of pressures down to $5 \times 10^{-11}$ Torr using magnetically suspended turbomolecular pumps (Osaka) and oil-free backing pumps (Anest Iwata)[45]. Phosphine and carbon dioxide gases were premixed and deposited at $2 \times 10^{-8}$ Torr via a glass capillary, while phosphine and water were deposited simultaneously using separate transfer mechanisms to avoid any pre-deposition reactions. Deposition continued until 750 nm of ice was deposited, which was measured in situ using laser interferometry by monitoring the interference fringes of a helium-neon laser (632.8 nm) that is reflected off the silver substrate[46]. The ice mixtures were found to be 10:1 water to phosphine and carbon dioxide to phosphine by using integrated infrared absorption coefficients[47]. The refractive index of the ice mixture, which is necessary for the thickness calculation, was determined to be the refractive index of the matrix ($n_{CO_2} = 1.27$[47], $n_{H_2O} = 1.29$[48]). The ices were irradiated with 5 keV electrons at a 70° angle to the surface normal for 1 h at currents of 100, 1000, and 5000 nA. Exploiting the CASINO simulation[49], the average penetration depth was found to be 285 nm ($H_2O/PH_3$) and 300 nm ($CO_2/PH_3$), while the maximum penetration depth was 650 nm ($H_2O/PH_3$) and 700 nm ($CO_2/PH_3$), which is less than the 750 nm ice thickness. The average dose was calculated to be $2.8 \pm 0.6$ eV per molecule ($CO_2/PH_3$) and $2.4 \pm 0.5$ eV per molecule ($H_2O/PH_3$) for the 100 nA irradiation, and these values scale linearly to 1000 and 5000 nA. The density of the ice mixture used for the CASINO simulation was the weighted average to the ice components: 0.90 g cm$^{-3}$ for PH$_3$[46,50], 0.94 g cm$^{-3}$ for H$_2$O[51], and 1.11 g cm$^{-3}$ for CO$_2$[47]. A Nicolet 6700 Fourier Transform Infrared Spectrometer monitored the ice during irradiation from 500 to 4000 cm$^{-1}$ with 4 cm$^{-1}$ resolution. Standard experiments utilized oxygen-18-labeled CO$_2$ and H$_2$O and were repeated at 5000 nA irradiation with non-isotopically labeled CO$_2$ and H$_2$O in order to confirm the infrared assignments, and both isotopologues are presented in Supplementary Tables 1 and 2.

**PI-ReTOF-MS**. After the irradiation, the ices were annealed at 1 K min$^{-1}$ while any subliming molecules were detected using a reflectron time-of-flight (ReTOF) mass spectrometer (Jordon TOF Products, Inc.) with single-photon ionization (PI)[45]. This photoionization process utilizes difference four wave mixing to produce vacuum ultraviolet light ($\omega_{vuv} = 2\omega_1 - \omega_2$). These experiments were performed with 10.86 eV photoionization energy and repeated at 9.93 eV to distinguish between the isomers of the phosphorus oxoacids. To produce 10.86 eV, the second harmonic (532 nm) of a pulsed neodymium-doped yttrium aluminum garnet laser (Nd:YAG, Spectra Physics, PRO-270, 30 Hz) was used to pump a Rhodamine 610/640 dye mixture (0.17/0.04 g L$^{-1}$ ethanol) to obtain 607 nm, which underwent a tripling process to achieve $\omega_1 = 202$ nm. A second Nd:YAG laser pumped an LDS 867 (0.15 g L$^{-1}$ ethanol) dye to obtain $\omega_2 = 890$ nm, which when combined with $2\omega_1$, using krypton as a non-linear medium, generated $\omega_{vuv} = 114$ nm (10.86 eV) at $10^{14}$ photons per pulse. The production of 9.93 eV occurred similarly except the second harmonic (532 nm) of the second Nd:YAG laser was used as $\omega_2$. The VUV light was spatially separated from other wavelengths using a lithium fluoride (LiF) biconvex lens (ISP Optics) and directed 1 mm above the sample to ionize subliming molecules. The ionized molecules

were mass analyzed with the ReTOF mass spectrometer where the arrival time to a multichannel plate is based on mass-to-charge ratios, and the signal was amplified with a fast preamplifier (Ortec 9305) and recorded with a bin width of 4 ns triggered at 30 Hz (Quantum Composers, 9518). The ReTOF recorded until the sample reached 300 K and held at constant temperature for 1 h, after which an infrared spectrum of the resulting residue was recorded.

**SIMS**. After each experiment, the wafer was removed and stored in an air-tight container under inert nitrogen atmosphere. The residue on the wafer was analyzed utilizing a time-of-flight secondary ion mass spectrometry instrument (TOF-SIMS 5-300, ION-TOF) equipped with a reflectron mass analyzer, a 30 keV $Bi_n$ liquid metal primary ion source (Bi-LMIG), and a low-energy electron flood gun. Spectra were acquired with $Bi_3^+$ in bunched mode with a focus of 2–3 mm, a beam current of 0.25 pA, and a total dose of $1.25 \times 10^8$ ions. Data were acquired from a 500 $\mu m^2$ analysis area. The positive and negative ions from the sputtered residue were recorded, and the residues were analyzed and compared with fragment assignments from calibrated phosphorus oxoacids adjusted for the mass shifts since our studies utilized $^{18}O$-labeled precursors.

**GC×GC−TOF-MS**. The gas chromatographic analysis was carried out by a GC×GC Pegasus IV D instrument coupled to a time-of-flight mass spectrometer ((TOF-MS), LECO Corp.). The TOF-MS system operated at a storage rate of 150 Hz, with a 25–700 amu mass range, a detector voltage of 1.5–1.8 kV, and a solvent delay of 12 min. The ion source and transfer temperatures were set to 503 K. The column set consisted of an Agilent J&W DB 5MS Ultra Inert primary column (30 m × 0.25 mm inner diameter (ID), 0.25 $\mu m$ film thickness) Press-Tight (Restek Corp.) connected to an Agilent DB Wax secondary column (1.40 m × 0.1 mm ID, 0.1 $\mu m$ film thickness). Helium was used as carrier gas at a constant flow rate $\bar{u} = 1$ mL min$^{-1}$. Sample volumes of 1 $\mu L$ were injected in the splitless mode into an ultra inert, single taper w/glass wool splitless inlet liner (Agilent) at an injector temperature of 503 K. All samples and reference compounds were analyzed with the identical temperature program. The primary oven was operated as follows: 313 K for 1 min, temperature increase of 5 K min$^{-1}$ to 473 K and held for 15 min. The secondary oven used the same temperature program with a constant temperature offset of 303 K. A modulation period of 5 s was applied for the liquid nitrogen cooled thermal modulator. Data were acquired and processed with LECO Corp. ChromaTOF software. Compound identification was performed by comparison with the chromatographic retention in both dimensions and mass spectra of authentic standards. The derivatization reagents including N,O-bis(trimethylsilyl) trifluoroacetamide (BSTFA) and trimethylsilyl chloride (TMCS), the internal standard methyl laurate, hexane, and water as well as phosphonic acid (HPO (OH)$_2$), phosphoric acid (H$_3$PO$_4$), pyrophosphoric acid (H$_4$P$_2$O$_7$), and sodium triphosphate (Na$_5$P$_3$O$_{10}$) were purchased from Sigma-Aldrich. Sample handling glassware was wrapped in aluminum foil and heated at 773 K for 3 h prior to usage to eliminate possible contamination. Eppendorf tips were sterile and the water used for extraction, standard solutions, reagent solutions, and blanks were of high-performance liquid chromatography grade. The residues were extracted with 10 × 50 $\mu L$ water from their silver wafers and transferred into conical reaction vials (1 mL V-Vial, Wheaton). The aqueous extracts were dried under a gentle stream of nitrogen and silylated with an excess of 50 $\mu L$ BSTFA and 10 $\mu L$ TMCS for 2 h at 353 K in the presence of the internal standard methyl laurate in hexane (5 $\mu L$, 10$^{-5}$ M). The derivatized mixtures were transferred into GC vials for their subsequent GC×GC−TOF-MS analysis. Procedural blanks were run in sequence to each sample in order to monitor background interferences.

Compound identification was performed by comparison with the chromatographic retention in both dimensions of authentic standards and mass spectra. Comparison with previously reported data on TMS derivatives of phosphorus oxoacids[31,43,44] let us identify multiple oxoacids. Supplementary Table 6 summarizes the analytes' retention times in the first and second chromatographic dimension as well as the corresponding mass spectra of the analyzed residue and standard. The mass spectra of all three phosphorus oxoacids are characterized by the molecular ion [M]$^+$ peak, the fragment ion [M-15]$^+$ formed by the loss of a methyl group as well as silicon-containing fragments at m/z 45 [CH$_3$SiH$_2$], 73 [(CH$_3$)$_3$Si], 75 [(CH$_3$)$_3$SiH$_2$], 133 [(CH$_3$)$_3$Si$_2$O$_2$], 147 [(CH$_3$)$_5$Si$_2$O], and 207 [(CH$_3$)$_5$Si$_3$O$_3$]. The phosphorus oxoacids detected in the residues were found to contain $^{18}O$ isotopes, whereas the standard samples used for identification were made of the natural isotopic composition dominated by $^{16}O$. Esterfication, necessary to increase the volatility of our target compounds in the gas chromatographic analysis, resulted in a shift of the tautomeric equilibria of phosphonic and phosphinic acid toward their more active pyramidal forms (Supplementary Fig. 1).

**Theoretical**. For the theoretical calculations, the geometries of neutral molecules and cations were optimized by the hybrid density functional B3LYP[52–55] with the cc-pVTZ basis set and thus obtained the harmonic frequencies. Their coupled cluster[56–59] CCSD(T)/cc-pVDZ, CCSD(T)/cc-pVTZ, and CCSD(T)/cc-pVQZ energies were then calculated and extrapolated to completed basis set limits[60], CCSD(T)/CBS, with B3LYP/cc-pVTZ zero-point energy corrections. The energies are accurate within 0.08 eV[61]. The adiabatic ionization energies were computed by

taking the energy difference between the ionic and the neutral states that correspond to similar conformation. At this level of theory, the adiabatic ionization energies are within 0.05 eV. The GAUSSIAN09 program[62] was utilized in the electronic structure calculations.

## Data availability
The data that support the findings of the current research are available from the corresponding author on request.

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

## Acknowledgements

This work was supported by NASA under Grant NNX16AD27G to The University of Hawaii. The Hawaii group would also like to thank the W.M. Keck Foundation to fund the surface science machine. This work has also been supported by the French government through the UCA$^{JEDI}$ Investments in the Future project managed by the National Research Agency (ANR) with the reference number ANR-15-IDEX-01. A.H.H. C., B.-J.S., and K.-H.C. thank the National Center for High-performance Computer in Taiwan for providing the computer resources utilized in the calculations.

## Author contributions

A.M.T., A.B., M.J.A., C.Z., and S.G. carried out the experiments; C.M. performed the gas chromatography analysis; B.-J.S., K.-H.C., and A.H.H.C. carried out the theoretical calculations; AM.T. analyzed the experimental data; A.M.T. and R.I.K. wrote the manuscript; and R.I.K. supervised the study.

## Additional information

**Competing interests:** The authors declare no competing interests.

