## [Peer Review File · Nature Communications]

Reviewers' comments:

Reviewer #1 (Remarks to the Author):

This manuscript describes the electron-irradiation of mixed PH₃-H₂O and PH₃-CO₂ ices at temperatures of 5 K and the subsequent analysis of the resulting reaction products using a variety of methods. Specifically, infrared spectroscopy, PI-ReTOF-Mass spectrometry, TOF-SIMS and GC methods were all employed in order to identify and build a connectivity pathways between the phosphorus-containing products.

The reasons behind performing the study were well laid out in the introduction. These centre on the well-recognised "phosphorus problem" in abiogenesis studies which states in essence that the most commonly encountered chemical form of phosphorus, phosphate salts (PO₄³⁻), are both highly water insoluble and very lower in chemical reactivity, thus making their involvement in the earliest stages of life emergence challenging from a chemical perspective (Gulick 1955). These observations, coupled with the recognition that contemporary biology uses phosphorus ubiquitously, has lead scientists to consider various ways in which Nature may have overcome the "phosphorus problem". Of greatest significance to the present study is phosphorus redox chemistry which provides routes to lower oxidation state phosphorus oxyacids of both greater water solubility and chemical reactivity. Since Gulick's prescient voice in the mid50's, others including Schwartz, Pasek and Kee for example, have pursued this redox model in various directions.

First and foremost, this current study is a most valuable one in that the authors have sought to explore an interstellar ice-irradiation simulation which, to the best of my knowledge, has not been used before. The experimental analytical work has been performed and reported both thoroughly and carefully. Therefore, on the technical, analytical and computational sides of the piece I do not have any issues. It would not be unreasonable however, to point out that the products observed in the current study are to be expected from the PH₃-oxygen source system as this is well-established P-chemistry (see: Fluck and Novobilsky: Fortschr. chem. Forsch. 1969, 13, 125).

I think the authors do a very solid job in laying the contextual foundation of their investigations within this context of P-redox chemistry. However, it is not made particularly clear to the reader that interstellar PH₃ is but one of a number of such IS P-based compounds which includes, CP, CCP, HCP, possibly PN and most commonly condensed Fe-Ni-P phases within iron meteorites (such as schreibersite). The question here then is whether there is something that particularly stands out with PH₃ within the context of this group of molecules or is one filling in the gaps in terms of irradiative behaviour?

The importance of the other small-molecule P-compounds are flagged in an earlier paper from this group (Turner et al. ApJ, 2016, 819, 97) but I notice that the reference list on the importance of schreibersitic phases in the production of lower oxidation state P-oxyacids has some notable omissions that I offer as being of some value to the authors (eg: Bryant and Kee, Direct evidence for the availability of reactive, water soluble phosphorus on the early Earth. H-Phosphinic acid from the Nantan meteorite. Chem. Commun., 2006, 2344-2346) which also produces oxyacids, including phosphite & hypophosphite, from a reduced oxidation state P-source. This paper exploits photochemical irradiation of the schreibersite-containing Nantan meteorite to afford P-oxyacids and whilst it is not mentioned in this paper per se, one key implication is that hydrolytic modification of schreibersite affords phosphine PH₃ (as per D. Glindemann, et al □Phosphine by bio-corrosion of phosphide-rich iron, □ Environmental Science & Pollution Research, 1998, 5, 71□74).

The authors also point out the curious case of the Murchison meteorite phosphonates as being of considerable relevance to the current study. I note that neither the current paper nor the Turner paper listed above reference a relevant contribution to this story: Gorrell, et al. On the origin of the Murchison meteorite phosphonates. Implications for pre-biotic chemistry. Chem. Commun.,

2006, 1643-1645.

Overall, a very thorough study on a significant interstellar P-source. The potential significance though as to how the flux of PH₃-containing ices impacting the early earth might compares to iron meteorite influx which has been estimated to bring some 10^{EXP8} kg_{yr}⁻¹ of reactive P in the form of schreibersite to the early earth during the putative LHB (M. A. Pasek & D. Lauretta, OLEB, 2008, 38, 5-21) are less clear.

Reviewer #2 (Remarks to the Author):

The discovery of an ice pathway from phosphine to phosphorous oxoacids is interesting and the presented analysis of the chemistry is convincing. I have no comments on this central part of the paper, beyond that it is perhaps somewhat too detailed for a nature communications main body text. As a result it is sometimes difficult to follow for a non-expert, but I expect that this will be addressed in more detail by the Nature editorial staff. Instead my main concern is the applicability of the presented study to interstellar and circumstellar ices, and therefore its relevance to phosphorous oxoacid abundances in meteorites and comets.

There needs to be some motivation as to why phosphine chemistry could be a major production pathway of phosphorous oxoacids in space. Are there e.g. theoretical arguments for why large amounts of PH₃ is expected in ice? To my knowledge, phosphine has only been detected in the envelope of a dying star, i.e. not in the dense interstellar medium where icy grain mantles grow and chemically evolve. Where detected, phosphine is a minor carrier of phosphorous (~2%) even if all phosphine survives the harsh diffuse interstellar environment and becomes incorporated into icy grain mantles the maximum H₂O:PH₃ ice ratio that could be achieved would be ~20,000:1 (using solar abundances of O and P and assuming that ~1/5 of the O is in H₂O and 2% of the P in PH₃). With so little PH₃ in the ice (and perhaps much less still), is there enough phosphine to produce meaningful abundances of the oxoacids through the proposed chemistry? How does it compare to other proposed production pathways?

Second, the mechanistic reasoning seems sensible as applied to the presented experiments. There is a lack of discussion, however, what the product branching ratios and formation efficiencies should be under interstellar conditions when the ice mixing ratio is orders of magnitude different and the electron / cosmic ray flux orders of magnitude lower. This needs to be addressed to claim applicability to such conditions.

Reviewer #1 (Remarks to the Author):

This manuscript describes the electron-irradiation of mixed PH₃-H₂O and PH₃-CO₂ ices at temperatures of 5 K and the subsequent analysis of the resulting reaction products using a variety of methods. Specifically, infrared spectroscopy, PI-ReTOF-Mass spectrometry, TOF-SIMS and GC methods were all employed in order to identify and build a connectivity pathways between the phosphorus-containing products. The reasons behind performing the study were well laid out in the introduction. These centre on the well-recognised "phosphorus problem" in abiogenesis studies which states in essence that the most commonly encountered chemical form of phosphorus, phosphate salts (PO₄³⁻), are both highly water insoluble and very lower in chemical reactivity, thus making their involvement in the earliest stages of life emergence challenging from a chemical perspective (Gulick 1955). These observations, coupled with the recognition that contemporary biology uses phosphorus ubiquitously, has lead scientists to consider various ways in which Nature may have overcome the "phosphorus problem". Of greatest significance to the present study is phosphorus redox chemistry which provides routes to lower oxidation state phosphorus oxyacids of both greater water solubility and chemical reactivity. Since Gulick's prescient voice in the mid50's, others including Schwartz, Pasek and Kee for example, have pursued this redox model in various directions.

We appreciate these comments and summary of the work that contextualize the motivation of the work and frames our study using a comparison of complementary studies about relevant phosphorus chemistry.

First and foremost, this current study is a most valuable one in that the authors have sought to explore an interstellar ice-irradiation simulation which, to the best of my knowledge, has not been used before. The experimental analytical work has been performed and reported both thoroughly and carefully. Therefore, on the technical, analytical and computational sides of the piece I do not have any issues.

We are proud to receive the reviewer's commendation about the quality and novelty of our work.

It would not be unreasonable however, to point out that the products observed in the current study are to be expected from the PH₃-oxygen source system as this is well-established P-chemistry (see: Fluck and Novobilsky: Fortschr. chem. Forsch. 1969, 13, 125).

While the formation of phosphorus oxoacids from the reaction of phosphine with carbon dioxide or water is consistent with previous literature—indeed, our experiments were designed to observe anticipated phosphorus oxoacids—our results contribute new knowledge by demonstrating the phosphorus chemistry in interstellar ice analogues, which the reviewer noted has not been used before, as opposed to studies utilizing non-ice chemistry such as those involving microwave discharge or aqueous solutions. Furthermore, our method provides isomeric information to

establish a better identity of expected interstellar products, which otherwise would require NMR analysis and thus be impractical for astrochemical ice studies.

I think the authors do a very solid job in laying the contextual foundation of their investigations within this context of P-redox chemistry. However, it is not made particularly clear to the reader that interstellar PH₃ is but one of a number of such IS P-based compounds which includes, CP, CCP, HCP, possibly PN and most commonly condensed Fe-Ni-P phases within iron meteorites (such as schreibersite). The question here then is whether there is something that particularly stands out with PH₃ within the context of this group of molecules or is one filling in the gaps in terms of irradiative behaviour?

Indeed, other phosphorus-containing compounds do exist, and we have added the other compounds that have been detected. The manuscript does already mention schreibersite as an iron-nickel phosphide that, along with phosphates, make up the bulk of phosphorus in iron meteorites. The exciting feature about these astrochemical phosphine experiments is that phosphine complements the previous gas-phase (CP, CCP, HCP, etc.) and mineral phase (iron-nickel phosphide) studies by investigating the phosphorus chemistry of ices, which has applicability toward both interstellar ice and cometary ice chemistry. Given that phosphine was first discovered and confirmed in the circumstellar/interstellar medium within the past 10 years, and that the phosphorus signal of comet 67P/Churyumov-Gerasimenko is attributed to phosphine, conditions are thus ripe for studies that explore astrochemical ice-based phosphorus with phosphine as the best candidate for a phosphorus-carrier.

The authors also point out the curious case of the Murchison meteorite phosphonates as being of considerable relevance to the current study. I note that neither the current paper nor the Turner paper listed above reference a relevant contribution to this story: Gorrell, et al. On the origin of the Murchison meteorite phosphonates. Implications for pre-biotic chemistry. Chem. Commun., 2006, 1643-1645.

We thank the reviewer for recommending this reference and have added it to our manuscript.

Overall, a very thorough study on a significant interstellar P-source. The potential significance though as to how the flux of PH₃-containing ices impacting the early earth might compares to iron meteorite influx which has been estimated to bring some 10^{EXP8} kgyr⁻¹ of reactive P in the form of schreibersite to the early earth during the putative LHB (M. A. Pasek & D. Lauretta, OLEB, 2008, 38, 5-21) are less clear.

This study would not hope to rival estimates about the ability of iron meteorites to deliver vast reservoirs of mineral phosphides to the early Earth, but instead demonstrate that alternative sources of extraterrestrial phosphorus are available that are capable of being building blocks for more complex biologically relevant compounds. While we are not suggesting a large quantity of phosphine-containing ices impacted Earth and provided the bulk of Earth's phosphorus, we are investigating if the products formed in these phosphine-containing ices could have delivered critical prebiotic compounds to Earth.

Reviewer #2 (Remarks to the Author):

The discovery of an ice pathway from phosphine to phosphorous oxoacids is interesting and the presented analysis of the chemistry is convincing. I have no comments on this central part of the paper, beyond that it is perhaps somewhat too detailed for a nature communications main body text. As a result it is sometimes difficult to follow for a non-expert, but I expect that this will be addressed in more detail by the Nature editorial staff. Instead my main concern is the applicability of the presented study to interstellar and circumstellar ices, and therefore its relevance to phosphorous oxoacid abundances in meteorites and comets.

We do appreciate the comment that the chemistry is convincing and hope the proceeding responses demonstrate the applicability of the study to interstellar ices.

There needs to be some motivation as to why phosphine chemistry could be a major production pathway of phosphorous oxoacids in space. Are there e.g. theoretical arguments for why large amounts of PH₃ is expected in ice? To my knowledge, phosphine has only been detected in the envelope of a dying star, i.e. not in the dense interstellar medium where icy grain mantles grow and chemically evolve. Where detected, phosphine is a minor carrier of phosphorous (~2%) even if all phosphine survives the harsh diffuse interstellar environment and becomes incorporated into icy grain mantles the maximum H₂O:PH₃ ice ratio that could be achieved would be ~20,000:1 (using solar abundances of O and P and assuming that ~1/5 of the O is in H₂O and 2% of the P in PH₃). With so little PH₃ in the ice (and perhaps much less still), is there enough phosphine to produce meaningful abundances of the oxoacids through the proposed chemistry? How does it compare to other proposed production pathways?

This appears to be the reviewer's primary concern and can be summarized that the reviewer worries the abundance of phosphine is not sufficiently adequate to be chemically meaningful. First, although phosphine's only interstellar detection has been in the circumstellar envelope of IRC+10216, where it represents a minor product, this detection and comparison of abundance is in the *gas* phase. The other phosphorus-containing molecules that compose the remainder of the phosphorus budget include CP, CCP, HCP, NCCP, and PN. Similarly, these are *gas* phase molecules that have limited use when discussing solid-phases. In fact, no phosphorus has been discovered in interstellar ices. This is due to the inability to use sensitive radio-astronomy to detect solid compounds, and thus fewer than a dozen ice-phase molecules have been detected with only ammonia and carbonyl sulfide (and possibly sulfur dioxide) containing an element outside the carbon-hydrogen-oxygen system. Thus, phosphine makes an excellent choice for a compound that has been depleted from the gas phase into the solid state. In fact, the assignment of the phosphorus signal in comet 67P/ Churyumov-Gerasimenko to phosphine means thus far phosphine is the only phosphorus-containing compound to be assigned in an interstellar or cometary ice. Ultimately however, the phosphorus composition of interstellar ices is *currently unknown*, and thus phosphine provides a reasonable source of phosphorus to model phosphorus chemistry in these ices.

Second, the mechanistic reasoning seems sensible as applied to the presented experiments. There is a lack of discussion, however, what the product branching ratios and formation efficiencies should be under interstellar conditions when the ice mixing ratio is orders of magnitude different and the electron / cosmic ray flux orders of magnitude lower. This needs to be addressed to claim applicability to such conditions.

One of the issues with this discussion is that it requires a significant amount of speculation about the phosphorus composition of the interstellar ices and there are no observations of these compounds. Our experiments demonstrate that phosphine molecules in water- or carbon dioxide-rich (i.e. oxygen rich) ices form oxidized phosphorus compounds (in particular H_3PO_4) and that phosphorus is the limiting atom. Thus, it is logical to expect H_3PO_4 as the major product in even more oxygen-rich ices. Also, as experiments cannot proceed for thousands of years, we have to accelerate the chemistries in the "laboratory ices". This is a well-accepted and validated approach in the laboratory astrophysics community. Our experiments used different doses for these ice mixtures as well with the product distribution in the IR spectra being very similar (nearly identical). Therefore, it is important to note that the laboratory simulation experiments try their best to replicate the conditions in space, but there are no perfect laboratory experiments accounting for distinct fluxes in space and in the lab. However, the dose-dependent and time-dependent studies as carried out in our laboratory, as well as other astrophysics laboratories, account for this.